# Experiment and Analysis of Submarine Landslide Model Caused by Elevated Pore Pressure

**Tao Liu [1,2,\*], Yueyue Lu [1], Lei Zhou [1], Xiuqing Yang [1] and Lei Guo [3,\*]**

[1] Shandong Provincial Key Laboratory of Marine Environment and Geological Engineering, Ocean University of China, Qingdao 266100, China; Lori_Lu123@163.com (Y.L.); zhoulei@stu.ouc.edu.cn (L.Z.); yxq19941108@163.com (X.Y.)

[2] Laboratory for Marine Geology, Qingdao National Laboratory for Maine Science and Technology, Qingdao 266061, China

[3] Institute of Marine Science and Technology, Shandong University, Qingdao 266000, China

\* Correspondence: ltmilan@ouc.edu.cn (T.L.); rendar_lx@163.com (L.G.)

**Abstract:** Hydrate decomposition is an important potential cause of marine geological disasters. It is of great significance to understand the dynamic relationship between hydrate reservoir system and the overlying seabed damage caused by its decomposition. The purpose of this study is to understand the instability and destruction mechanisms of a hydrated seabed using physical simulations and to discuss the effects of different geological conditions on seabed stability. By applying pressurized gas to the low permeability silt layer, the excess pore pressure caused by the decomposition of hydrate is simulated and the physical appearance process of the overlying seabed damage is monitored. According to the test results, two conclusions were drawn in this study: (1) Under the action of excess pore pressure caused by hydrate decomposition, typical phenomena of overlying seabed damage include pockmark deformation and shear–slip failure. In shallower or steeper strata, shear-slip failure occurs in the slope. The existence of initial crack in the stratum is the main trigger cause. In thicker formations or gentler slopes, the surface of the seabed has a collapse deformation feature. The occurrence of cracks in the deep soil layer is the main failure mechanism. (2) It was determined that the thickness and slope of the seabed, among other factors, affect the type and extent of seabed damage.

**Keywords:** hydrate decomposition; slope stability; excess pore pressure; failure mechanism

## 1. Introduction

Gas hydrates are special ice-like compounds; its stability is affected by temperature and pressure [1]. Therefore, the pressure of the stable zone decreases or temperature rises (caused by activities such as earthquakes, volcanoes, climate change, or a drop in sea level), leading to hydrate dissociation [2]. Thermal disturbances to a hydrate stratum during hydrate exploitation or natural environmental changes may cause dissociation of the hydrate. And the dissociation of the hydrate releases 164 times the volume of gas and 0.8 times the volume of water into the pore space [3].

This compound is not only a natural source of clean energy but also a potential factor that induces geological hazards [4]. Hydrate decomposition and submarine landslides are indeed related, and hydrates may be an important reason for associated submarine landslides [5–7]. Figure 1 [8] shows that the area where the submarine landslide occurs contains hydrate deposits, which indirectly indicates that hydrates are associated with submarine landslides. Hydrates may not always cause submarine landslides on their own [9]; however, when combined with other factors such as earthquakes, sediment subsidence, or sudden gas eruptions under the fault, they may cause the stability of the seabed slope to be disrupted [10,11]. Despite the observations of ongoing and previous studies, the mechanism of hydrate decomposition in slope sediments is still unclear and requires further investigation.

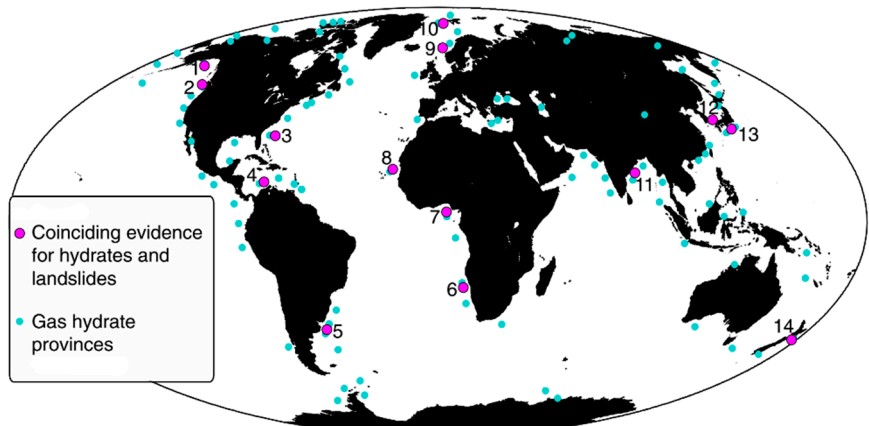

**Figure 1.** Global compilation of large submarine landslides in areas with gas hydrates [8].

Hydrate, as the influencing factor of geological hazard, has two main effects on strata: one is its effect on the strength of sediments; the other is the increase of pore pressure caused by hydrate decomposition. Yelisetti et al. [4] observed that the contrast in sediment strength between hydrated and non-hydrated sediments acts as a glide plane for failure on the northern Cascadia margin. A lot of experimental research and theoretical analysis have been carried out on the influence of hydrate on sediment strength. [12–15] On the one hand, the decomposition of hydrate will lead to the decrease of sediment strength [16] On the other hand, the cementation of hydrates in the sediment enhances the mechanical properties of the sediment and is influenced by factors such as type and saturation of the hydrate [17,18], skeleton and structural properties of the sediment [19–21], and distribution of the hydrate [22,23].

Current research indicates that the main mechanism of seabed instability is the high pore pressure caused by hydrate decomposition. In order to elucidate the deformation process and failure characteristics of the soil under hydrate decomposition, a series of experimental studies were performed by simulating hydrate decomposition. Wei [24] determined that gas pressure and cap thickness have an effect on soil morphology damage. Zhang [25,26] and Liu et al. [27] simulated the destruction of sediments caused by thermal decomposition of tetrahydrofuran hydrate. The results revealed that the formation has a slow slump or stratified fissure failure and even significant destruction of soil eruption. The centrifugal model test can simulate a real stress field. Zhang et al. [28] simulated submarine landslide and soil flow trends under excess pore pressure using this test. Zhang [29] simulated the submarine landslide caused by the thermal decomposition of hydrate in the centrifuge and identified the basic phenomena involved in the process. Acosta et al. [30] studied the effects of seafloor soil composition and water content on submarine landslides using the centrifuge test. It is therefore apparent that physical model tests play an important role in the understanding of submarine landslides and in elucidation of the influencing factors. [31] However, relevant experimental research in this area is deficient with respect to the study of the physical evolution process of submarine landslides, especially a detailed study of the seabed failure mode and critical failure state.

The purpose of this paper is to study the physical evolution process of submarine landslide through a physical simulation experiment and to understand the dynamic relationship between the hydrate reservoir system and destruction of the overlying seabed caused by its decomposition. According to the failure phenomenon of hydrate seabed instability, the mechanism of submarine landslide is analyzed, and the effect of different geological conditions on seabed stability is discussed.

## 2. Experimental Models

### 2.1. Experimental Introduction

In this paper, the thickness and slope of the overlying seabed were considered the main control variables in this investigation. Moreover, nine sets of seabed models with different stratigraphy conditions were established to simulate the instability process of the overlying seabed caused by hydrate decomposition. Considering the size of the model box, the thickness of the overlying seabed was set to 20 cm, 13 cm, and 8 cm in decreasing order of thickness. In addition, the slope of the seabed was set to 5°, 10°, and 15°. Table 1 shows the experimental settings for the nine seabed models.

**Table 1.** The parameters used for the test groups.

| Experiment Number | Seabed Thickness/D (cm) | Seabed Slope/$\alpha$ (°) | Water Height/h (cm) | Sand Layer Thickness/s (cm) |
|---|---|---|---|---|
| 20-5 | 20 | 5 | 35 | 2 |
| 20-10 | 20 | 10 | 36 | 2 |
| 20-15 | 20 | 15 | 44 | 2 |
| 13-5 | 13 | 5 | 35 | 2 |
| 13-10 | 13 | 10 | 36 | 2 |
| 13-15 | 13 | 15 | 36 | 2 |
| 8-5 | 8 | 5 | 35 | 2 |
| 8-10 | 8 | 10 | 36 | 2 |
| 8-15 | 8 | 15 | 36 | 2 |

The thickness of the seabed was controlled by the volume of soil. The slope of the seabed was controlled using a jack placed under the model box. To prevent direct disturbance of the bottom sedimentary soil by the gas, a layer of sandy soil with high permeability was deposited on the bottom plate of the gas application system up to a thickness of 2 cm.

### 2.2. Experimental Device Design

In order to simulate the hydrate decomposition environment and to monitor seabed damage, a visual observation device was designed to simulate the slope failure process. The device consists of an acrylic model box with a controllable slope, a gas application system, and data and image acquisition systems. The overall layout of the test device is shown in Figure 2.

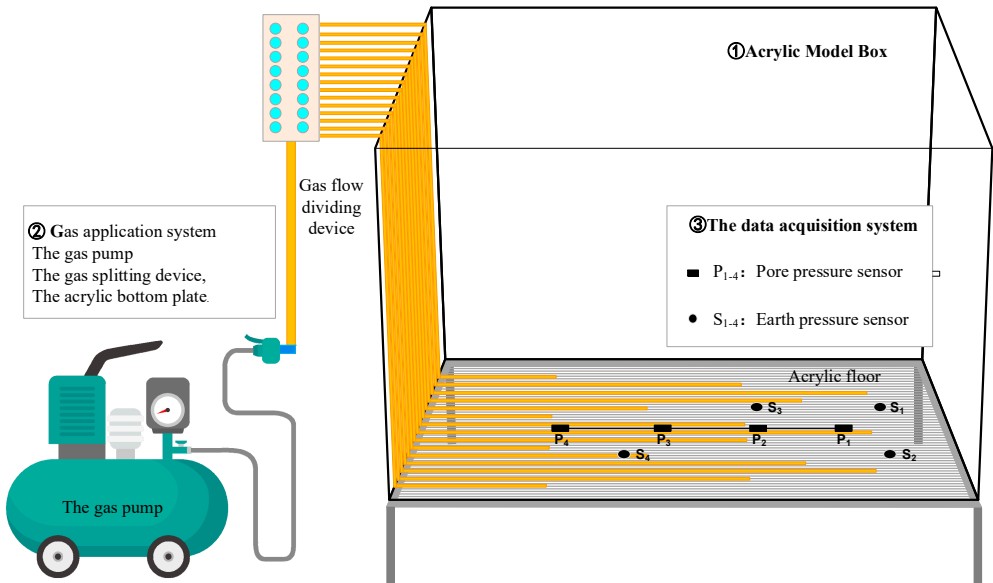

**Figure 2.** Schematic diagram of the test device.

### 2.2.1. Acrylic Model Box

The acrylic model box consists of a polymethyl methacrylate (pmma) sink and a tiltable bracket. As shown in Figure 3, the dimensions of the pmma sink are 80 cm (L) × 40 cm (W) × 50 cm (H) and the thickness of the glass is 2 cm. The tiltable bracket controls the tilt angle of the sink, which can be adjusted from 0° to 25°, simulating the seabed slope at different angles.

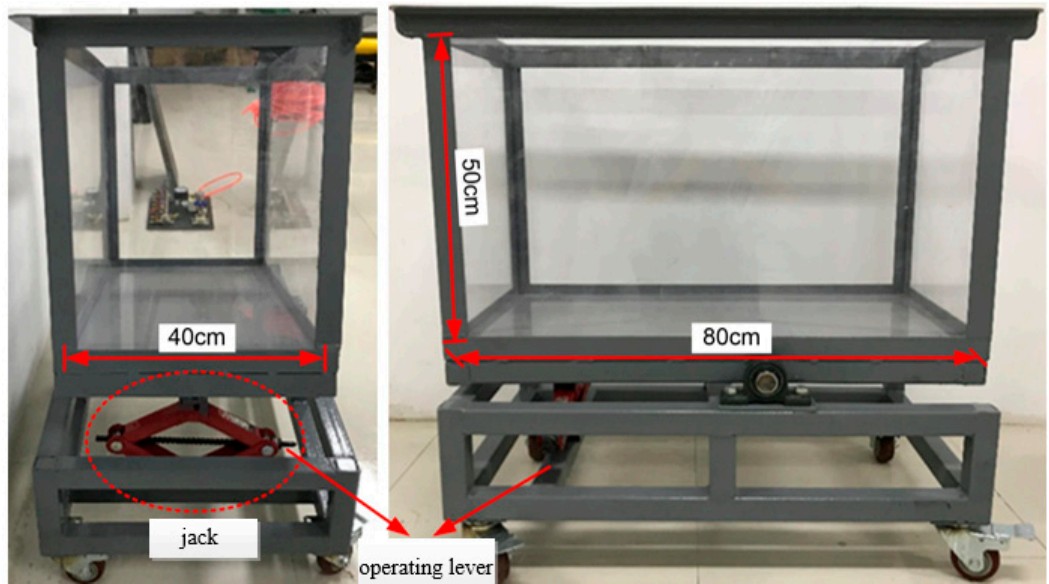

**Figure 3.** Acrylic model box used in the tests.

### 2.2.2. Gas application System

Figure 4 shows a gas application system consisting of a gas pump, a gas splitting device, and an acrylic bottom plate. The model of the air pump consists of a 550 W × 3 air compressor that can supply a maximum gas source pressure of 0.8 MPa. The gas splitting device is used to adjust gas pressure in the range of 0 kPa to 40 kPa. There are 15 grooves on the surface of the acrylic bottom plate to fix the ventilation hose. Uniform air distribution was used for the experiments, and the layout of the ventilation duct is shown in Figure 4.

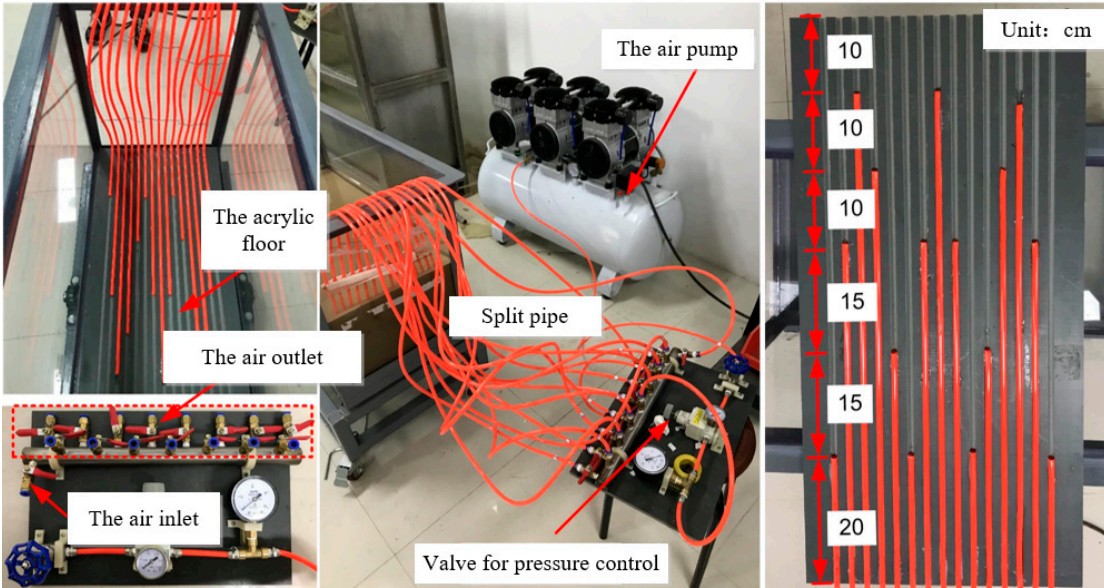

**Figure 4.** Gas application system for tests.

### 2.2.3. Data Acquisition System

The data acquisition system consists of a pore pressure sensor, an earth pressure sensor, data acquisition instrumentation, and data analysis software. In the experiments, four pore pressure sensors and four earth pressure sensors are used, as shown in Figure 2. The sensors were connected to the DEVE-43A eight-channel data acquisition instrument (see Figure 5c) using DEVESoftX software to read the data.

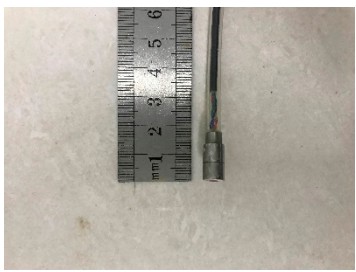

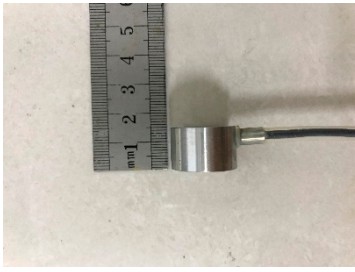

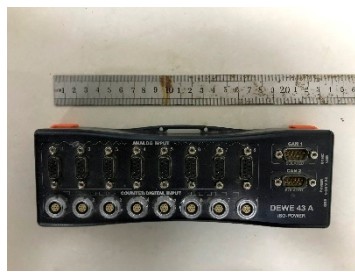

(**a**) Model-6007 pore pressure sensor    (**b**) YSV3211 earth pressure sensor    (**c**) GECO-16A data acquisition instrument

**Figure 5.** Sensors used in the tests. (**a**) Model-6007 pore pressure sensor; (**b**) YSV3211 earth pressure sensor; (**c**) GECO-16A data acquisition instrument.

### 2.3. Description of Model Tests

### 2.3.1. Experimental Materials

In this experiment, the silty soil of the Yellow River Delta beach was used. After pretreatment, basic parameters such as moisture content ($\omega$), density ($\rho$), dry density ($\rho_d$), and specific gravity ($G_s$) of the soil samples were measured. Table 2 shows a summary of the main parameters of the silt.

**Table 2.** Summary of the parameters of the test soil.

| Properties Parameters | | Value |
|---|---|---|
| Moisture content ($\omega$) | | 28.3% |
| Density ($\rho$) | | 2.0 g/cm$^3$ |
| Dry density ($\rho_d$) | | 1.55 g/cm$^3$ |
| Specific gravity ($G_s$) | | 2.7 |
| Void ratio ($e$) | | 0.75 |
| Porosity ($n$) | | 43% |
| Plasticity index ($I_p$) | | 6.9 |
| Saturated unit weight ($\gamma_s$) | | 16.0 kN/m$^3$ |
| shear strength of uu test | Cohesion ($c$) | 7 kPa |
| | Internal friction angle ($\varphi$) | 20° |

The salt content of seawater has a significant influence on the state and strength of coastal sedimentary soil [32]. The density of the sediment deposited in simulated seawater with 3.5% NaCl content is significantly lower than that of soil consolidated in fresh water. The soil had a small viscosity coefficient and small pore size, which are consistent with sedimentary soil under real sea conditions. Therefore, an aqueous solution with 3.5% NaCl was used in this experiment to simulate real seawater.

### 2.3.2. Experimental Procedure

There are three key issues in this experimental investigation. The first is accurate simulation of the excess pore pressure environment caused by hydrate decomposition under laboratory conditions. The second is observation of the physical evolution process of the overlying seabed damage. The third

is the monitoring of changes in pore pressure and earth pressure in the sedimentary layer. Therefore, ensuring the consolidation quality of the deposited layer and preventing dissipation of excess pore pressure are key aspects to the success of this investigation.

Setting up the Experimental Device

To prevent local damage caused by local application of gas, 15 ventilated rubber hoses were equally divided into three groups, as shown in Figure 6. These hoses were evenly fixed into the grooves at the bottom of the acrylic plate to achieve uniform gas distribution. The 15 rubber hoses were externally connected to the pressure control valve and ultimately, to the air source preparation machine installed next to the model box. After the gas application system was connected, the acrylic bottom plate with the rubber hose was placed at the bottom of the model box.

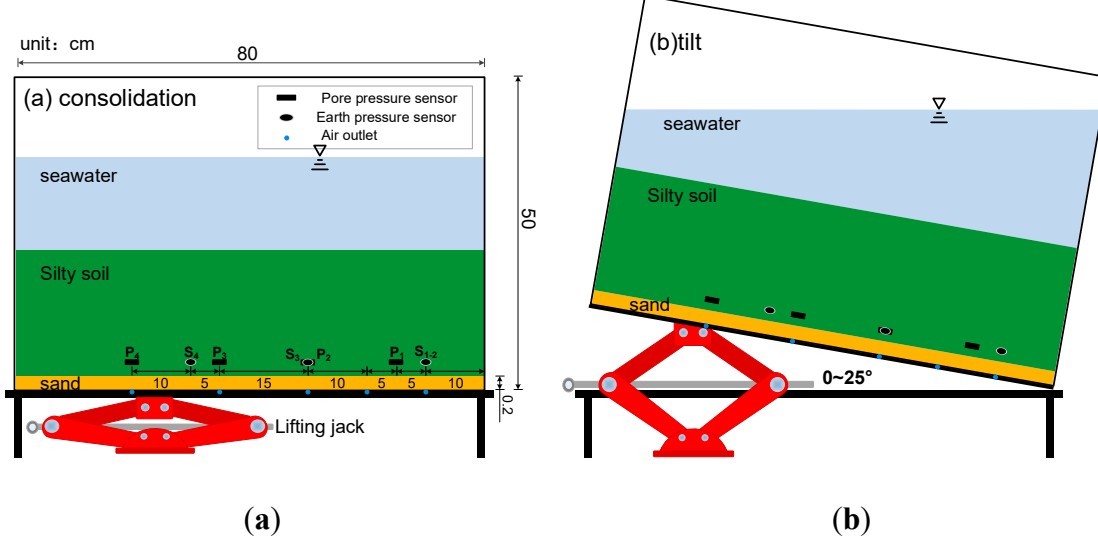

(**a**)                                        (**b**)

**Figure 6.** Schematic diagram of sedimentary soil sample preparation. (**a**) consolidation; (**b**) tilt.

Setting up the Sensors

Figure 6 shows the layout details of the sensors in the model box. Four pore pressure sensors are distributed along the longitudinal axis of the model box, and P1 is located near the slope. Two earth pressure sensors (S1 and S2) are evenly installed at the slope position, and the other two earth pressure sensors (S3 and S4) are distributed along the longitudinal axis of the model box.

Preparation of SEABED

To prevent direct disturbance of the bottom sediment during gas application, a 2-cm layer of sand with high permeability was first placed at the bottom of the model box to buffer the airflow impact. The treated silt soil sample was thoroughly mixed with simulated seawater to prepare a soil sample with water content of 60%. The soil sample was then stirred using a blender, and the uniformity of the prepared soil bed was ensured while stirring rapidly. Mixing of the soil sample with a blender ensures homogeneity of the sample while stirring quickly. Subsequently, the soil samples were transferred to the sink in batches. The simulated seawater was then slowly introduced into the model box. Finally, the soil sample was naturally consolidated for 7 days.

Data Acquisition

The camera was placed on the side of the model box and the height was adjusted to be flush with the model. The sensors were then connected to the DEVE-43A eight-channel data acquisition instrument to begin the process of data collection and the camera was powered on. When the data

were stable, the model box was rotated to the experimental angle to start the experiment. As shown in Figure 6, the soil layer may slip slightly due to the change of the seabed slope. After the data monitored by the sensors were stabilized, timing was initiated and subsequent experiments were performed. The gas source preparation machine was then opened to generate high-pressure gas. In addition, the control valve of the gas distribution device was opened. Gas was then pumped into the model box at a fixed step size of 2 kPa per 30 s. Changes in the soil layers and data fluctuations of the pore pressure sensors were recorded. The test was terminated when significant damage occurred in the local layer (e.g., the entire slope collapsed). The slope deformation was monitored by a camera mounted next to the model box.

At the end of the experiment, the powers for the gas application system and camera were turned off and the water in the box was siphoned to record the damage of the soil layer. In addition, the number, length, and width of the surface collapse and cracks of the seabed were measured, and the experimental data were compiled.

## 3. Experimental Results and Phenomena

In the process of experiment, the evolution process of formation instability and failure is recorded in real time by the camera. At the same time, at the end of each experiment, the number, diameter and width of annular collapse, vertical crack and sliding soil mass on the slope surface were measured to quantify the degree of seabed surface damage.

Table 3 lists the detailed information of the types and extent of seabed surface damage in the nine groups of experiments. It can be seen from the table that the destruction of the overlying seabed caused by hydrate decomposition simulated in this paper can be divided into two failure modes. One is the sediment collapse deformation like pockmark, which occurs in the thicker or slower strata—the destruction phenomenon is relatively mild, such as model 20-5, 20-10, 20-15 and 13-5. The other is the shear-slip failure, which occurs on the shallow and steep strata, such as model 13-15, 13-10, 8-5 and 8-10. In the 8-15 model, the slope is larger, the overlying soil layer is thinner, and the failure phenomenon is intense, which is not discussed in this paper.

**Table 3.** Detailed information on the mode and degree of slope failure.

| Number | Thickness/$D$ (cm) | Angle/$\alpha$ (°) | Failure Mode | Failure Time (s) | Failure Level (cm) |
|--------|--------------------|--------------------|--------------|------------------|---------------------|
| 20-5 | 20 | 5 | Pockmark | 420 | 2; Diameter: $8 \times 12.5$, $9.6 \times 6$ |
| 20-10 | 20 | 10 | Pockmark | 380 | 1; Diameter: $29 \times 25.3$ |
| 20-15 | 20 | 15 | Pockmark | 335 | 1; Diameter: $40 \times 35.7$ |
| 13-5 | 13 | 5 | Pockmark | 290 | 2; Diameter: $24 \times 20.3$, $21.5 \times 10.2$ |
| 13-10 | 13 | 10 | Shear–slip failure | 190 | 2; Width: 5; Slip zone: 27 |
| 13-15 | 13 | 15 | Shear–slip failure | 150 | 1; Width: 9.5; Slip zone: 50 |
| 8-5 | 8 | 5 | Shear–slip failure | 90 | 2; Width: 4.3, 5; Slip zone: 59 |
| 8-10 | 8 | 10 | Shear–slip failure | 80 | 5; Width: 3.5; Slip zone: 62 |
| 8-15 | 8 | 15 | Eruption failure | 20 | Overall liquefaction |

### 3.1. Collapse Deformation

Figures 7–10 represent the images and profiles of the initial surface, angle-adjusted initial sliding surface, and main failure surface of models 20-5, 20-10, 20-15 and 13-5, respectively. As a result of the adjustment of the angle of the model box before the start of the experiment, a slight initial slip occurred in the sediment layer. There was also a slight deformation of the surface of the slope. Figure 9 shows that some soils on the initial sliding surface have a distinct sliding displacement, due to the high steepness of the slope, especially in the 20-15 model. After the start of the experiment, with the gradual application of pressurized gas, the pore pressure in the soil layer gradually increased. When the pore pressure was allowed to accumulate for a period of time, the pore pressure was suddenly

released, triggering the main damage of the seabed. It should be noted that although the slope angles of the three models are different, identical failure mechanisms were observed as pockmarks on the slope's surfaces.

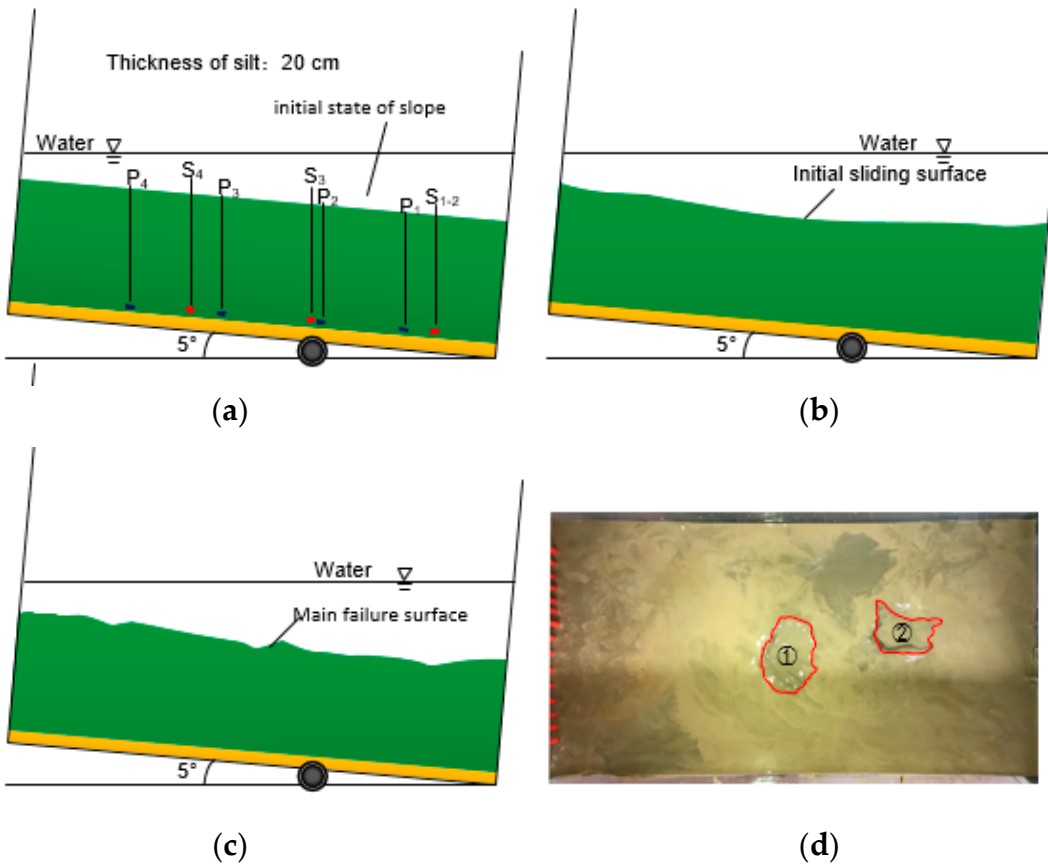

**Figure 7.** Profile and plan views of the slope after failure in Model 20-5. (**a**) Before Angle adjustment; (**b**) After the Angle adjustment; (**c**) After the destruction; (**d**) top view of end of experiment.

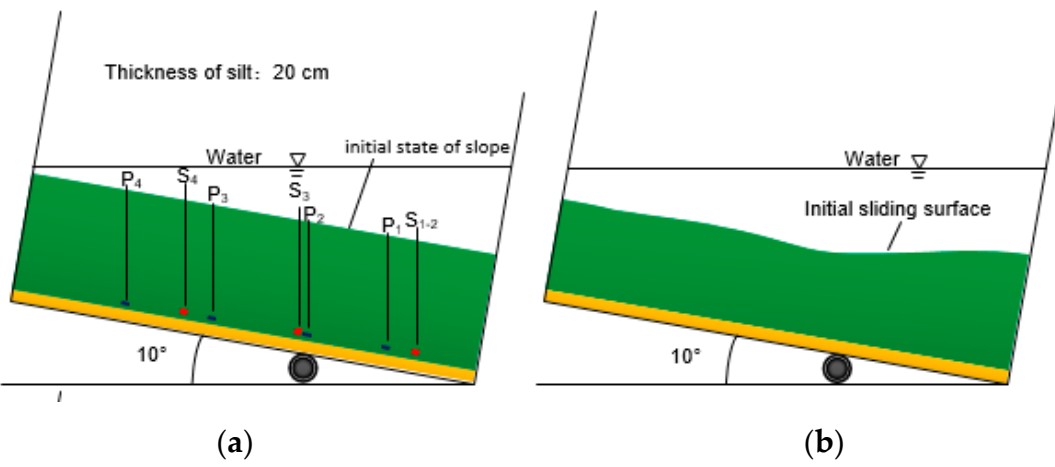

**Figure 8.** *Cont.*

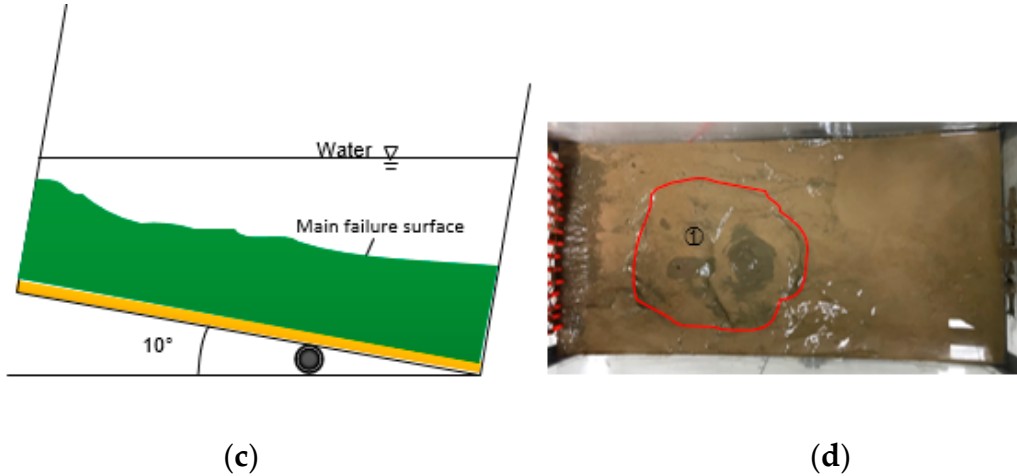

(c)                                        (d)

**Figure 8.** Profile and plan views of the slope after failure in Model 20-10. (**a**) Before Angle adjustment; (**b**) After the Angle adjustment; (**c**) After the destruction; (**d**) top view of end of experiment.

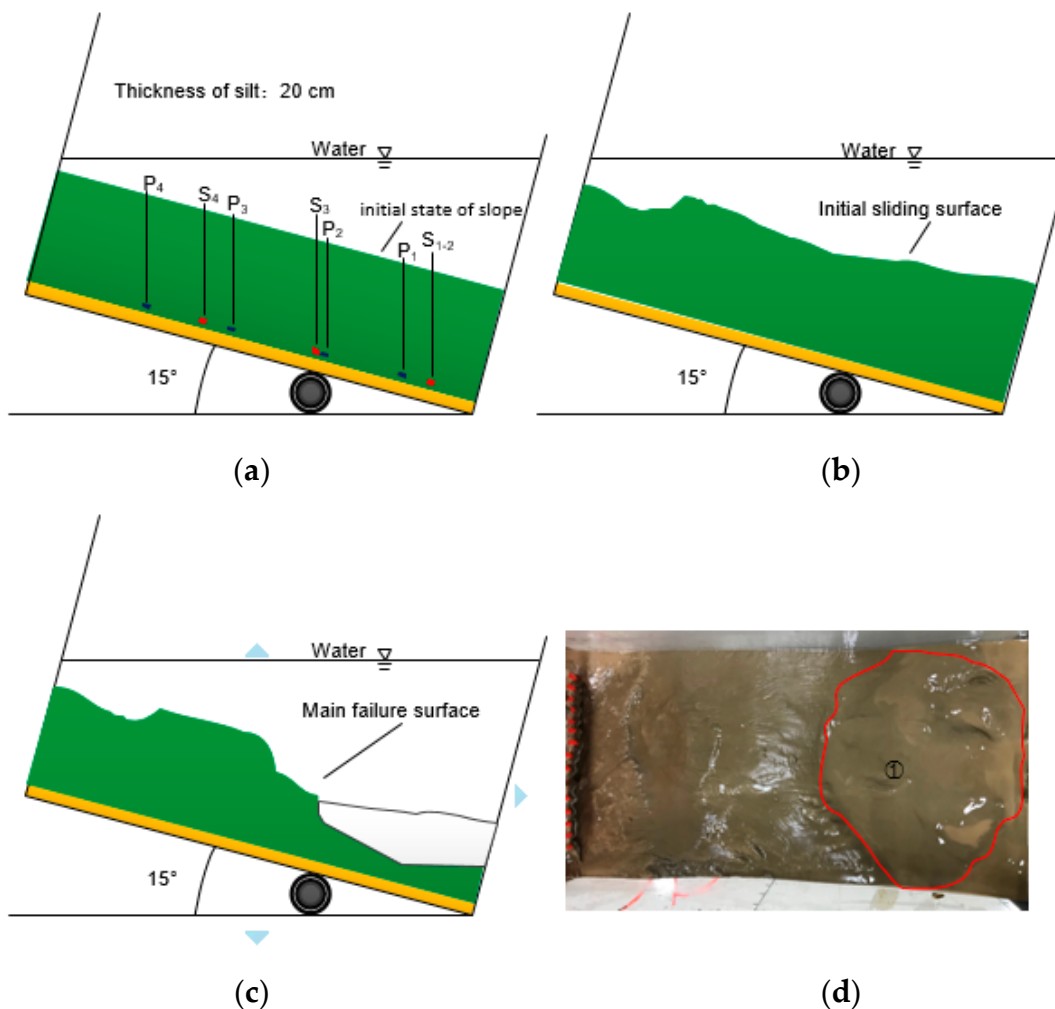

(a)                                        (b)

(c)                                        (d)

**Figure 9.** Profile and plan views of the slope after failure in Model 20-15. (**a**) Before Angle adjustment; (**b**) After the Angle adjustment; (**c**) After the destruction; (**d**) top view of end of experiment.

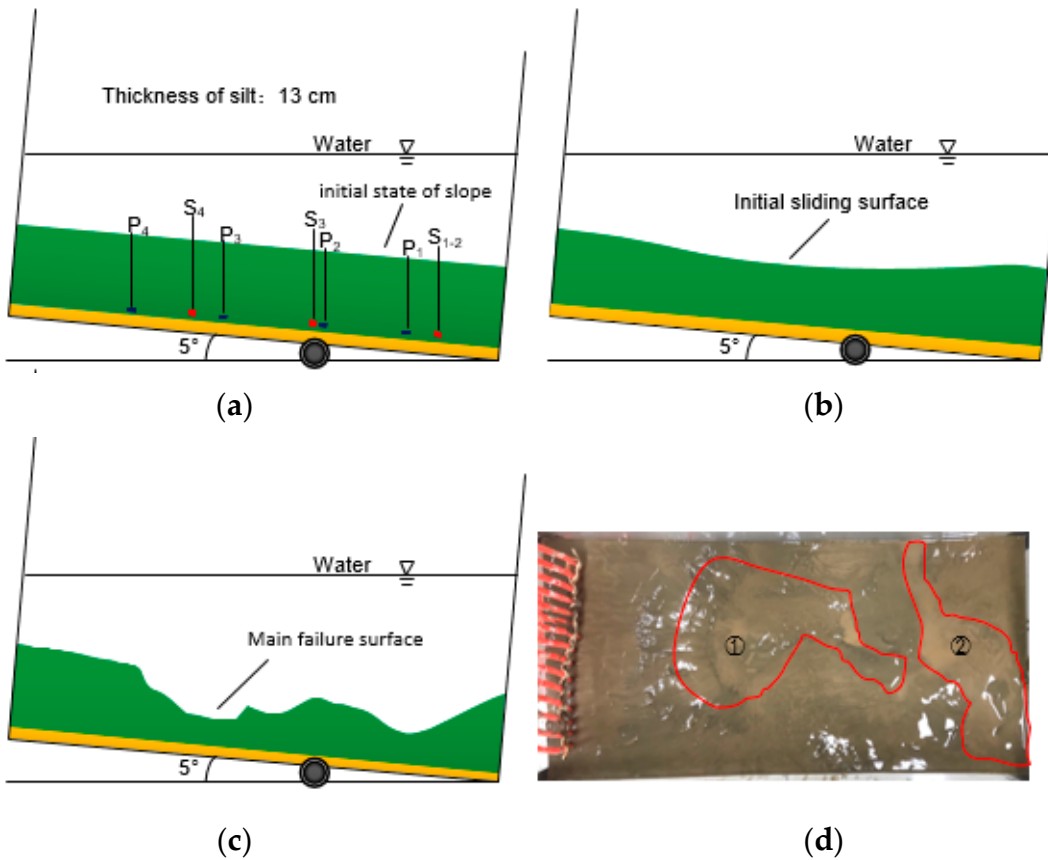

**Figure 10.** Profile and plan views of the slope after failure in Model 13-5. (**a**) Before Angle adjustment; (**b**) After the Angle adjustment; (**c**) After the destruction; (**d**) top view of end of experiment.

It is worth noting that in the 13-5 model (as shown in Figure 10), as the thickness of the soil layer decreases, the stress of the overlying soil layer decreases, and the gas accumulated at the bottom is more likely to cause damage. Therefore, in this model, the seabed surface has a larger extent of damage, and the collapse deformation zone covers two-third of the entire seabed surface. Moreover, the soil in the collapsed area exhibits relatively high fluidity.

### 3.2. Shear-Slip Failure

Figures 11–14 show the slope failure diagrams for the 13-10, 13-15, 8-5 and 8-10 models, respectively. Crack damage through the silt layer can be observed in the 13-10, 13-15, 8-5 and 8-10 models compared to the collapse deformation, especially in the 13-15 model. After the initial angle adjustment, the slope had distinct vertical cracks under the influence of its weight. These cracks destroy the soil structure and reduce the effective stress. In addition, they also provide a channel for the release of excess pore pressure. After the gas is applied, it is more likely to accumulate at the crack. Under the action of gas pressure, the silt at the crack shows a clear tendency to move toward the foot of the slope.

Moreover, owing to the incompatibility of sedimentary soil movement, the crack gradually develops. In addition, in the 13-10 and 13-15 models, the seabed surface exhibited obvious slip along the crack. The silt at the foot of the slope was highly mixed with water and no longer intact. The soil had strong fluidity and was in a liquefied state. In both models, the main cause of slippage of the slope may be the vertical cracks that exist inside the slope. The high excess pore pressure created by the accumulated gas causes an upward seepage pressure that promotes the movement of the water–soil mixture up the fracture, resulting in a weak sliding surface inside the soil. Eventually, the soil layer undergoes shear failure along the sliding surface. Figures 11 and 12 also show that the main sliding surfaces of the slopes occur at the slight deformations caused by initial sliding.

In the 8-10 and 8-15 models, silt particles are highly mixed with water and exhibit high liquefaction characteristics. This is because in this shallow seabed, the deposited soil is more likely to slide downwards owing to the dual effects of gravity and high pore pressure. Incompatible deformation, which occurs during the sliding of the sedimentary soil, results in the formation of cracks in the soil. The tiny silt particles carried by the mixed fluid, drift upwards through the cracks, causing the soil to soften, slip, and liquefy.

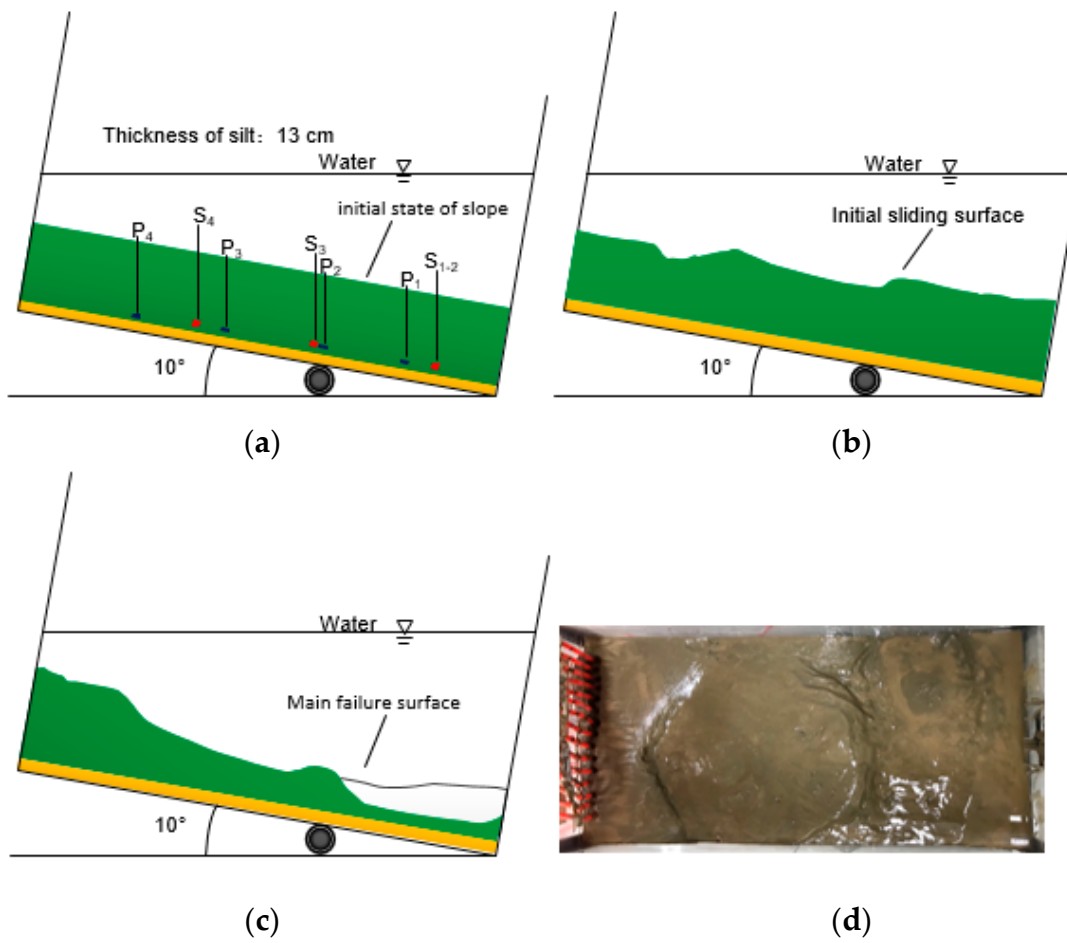

**Figure 11.** Profile and plan views of the slope after failure in Model 13-10. (**a**) Before Angle adjustment; (**b**) After the Angle adjustment; (**c**) After the destruction; (**d**) top view of end of experiment.

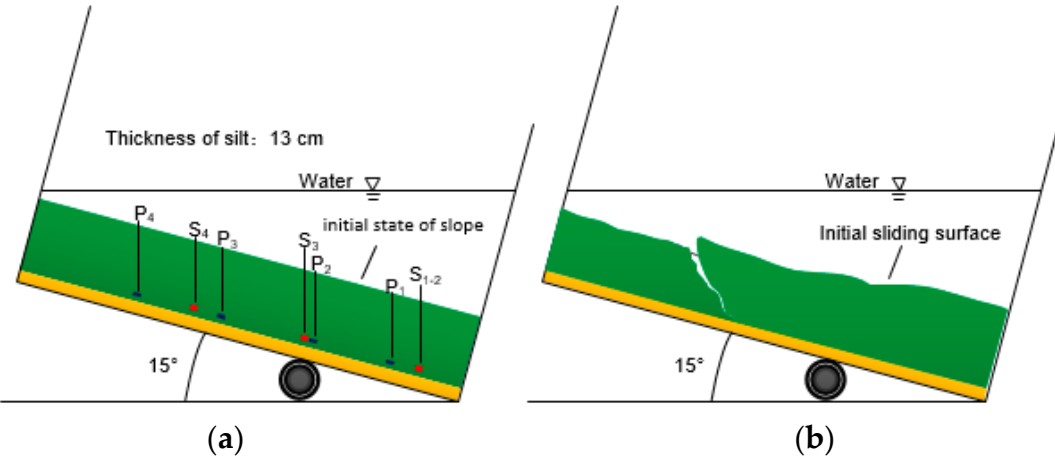

**Figure 12.** *Cont.*

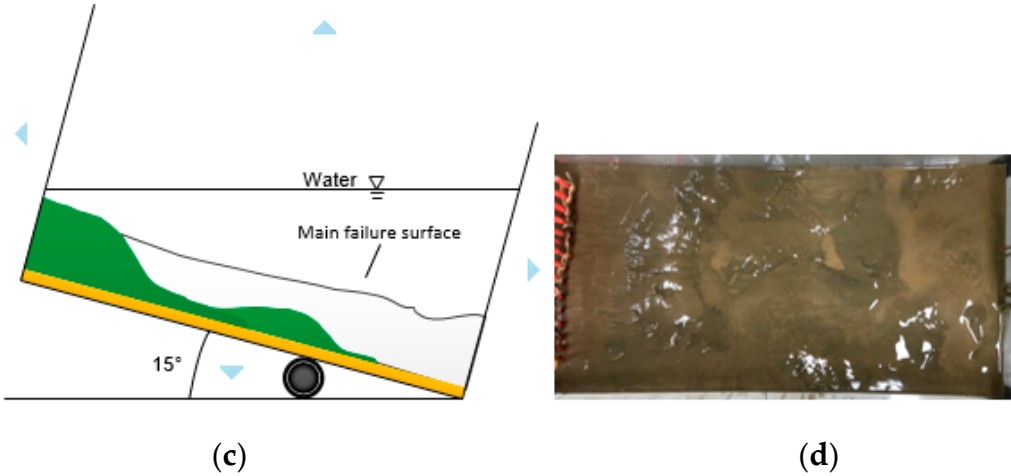

**(c)**　　　　　　　　　　　　　　　　　　　　**(d)**

**Figure 12.** Profile and plan views of the slope after failure in Model 13-15. (**a**) Before Angle adjustment; (**b**) After the Angle adjustment; (**c**) After the destruction; (**d**) top view of end of experiment.

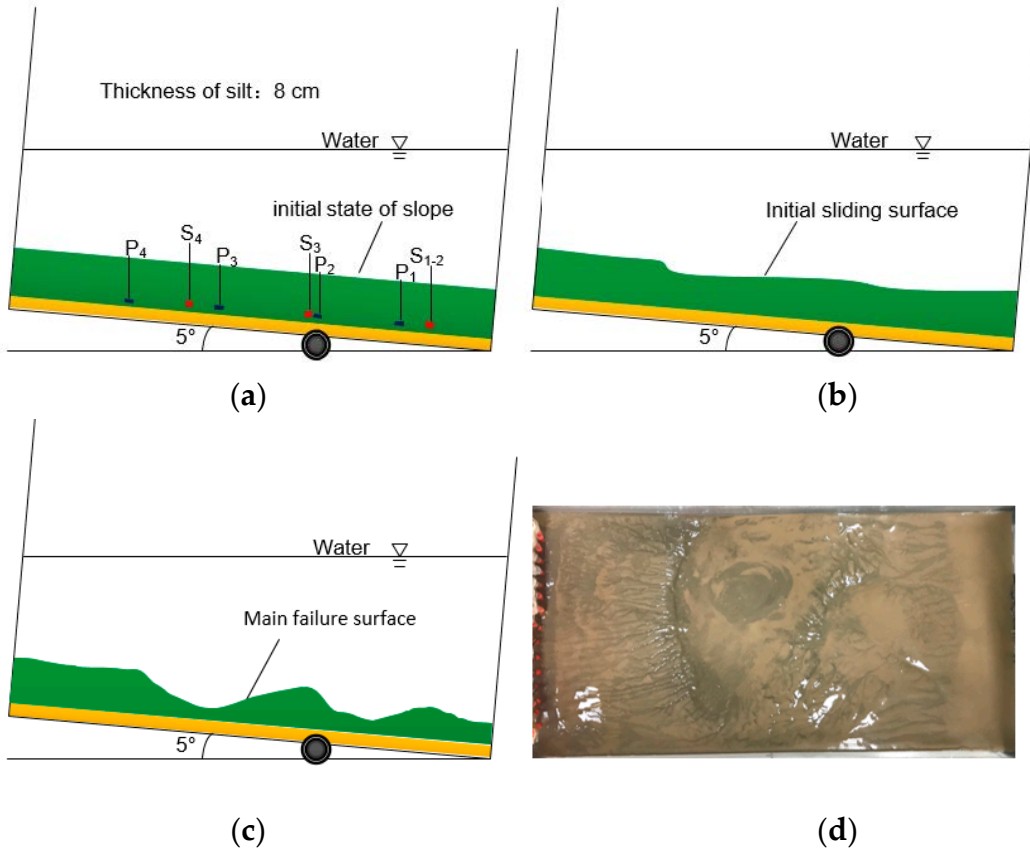

**(a)**　　　　　　　　　　　　　　　　　　　　**(b)**

**(c)**　　　　　　　　　　　　　　　　　　　　**(d)**

**Figure 13.** Profile and plan views of the slope after failure in Model 8-5. (**a**) Before Angle adjustment; (**b**) After the Angle adjustment; (**c**) After the destruction; (**d**) top view of end of experiment.

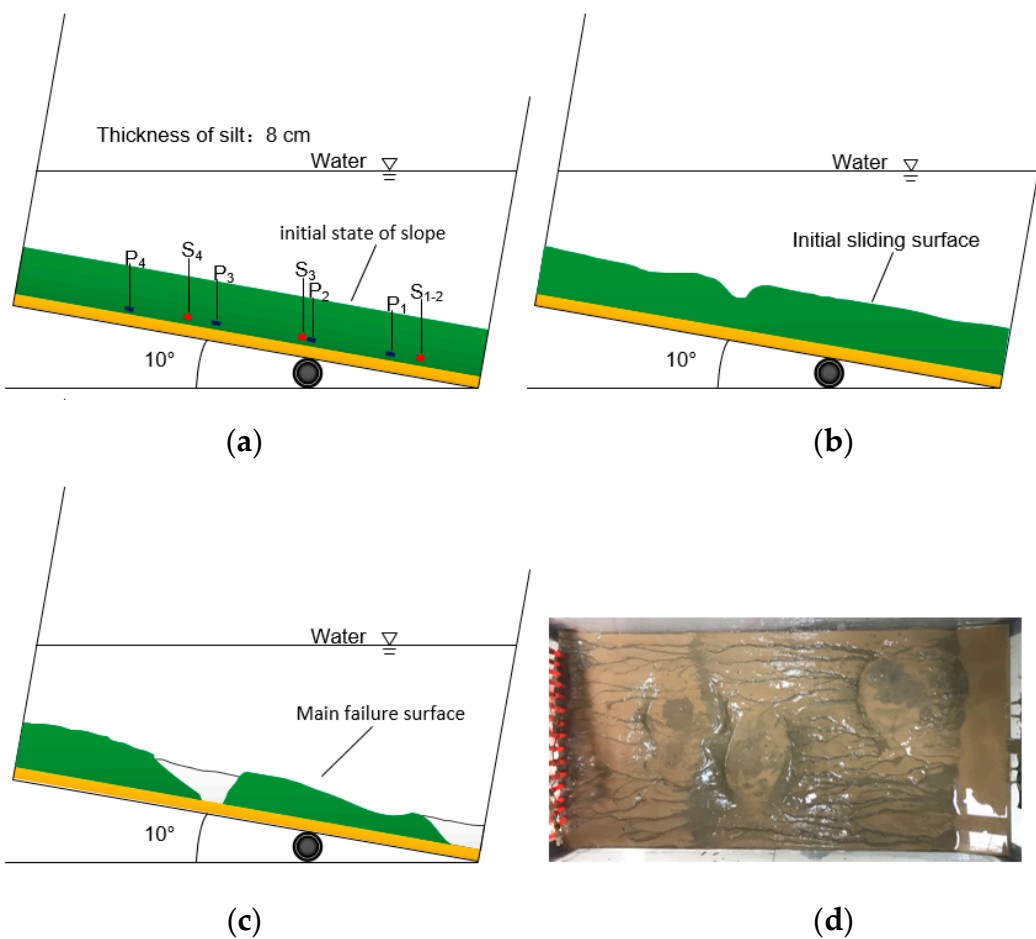

**Figure 14.** Profile and plan views of the slope after failure in Model 8-10. (**a**) Before Angle adjustment; (**b**) After the Angle adjustment; (**c**) After the destruction; (**d**) top view of end of experiment.

## 4. Discussion of Experimental Phenomena

### 4.1. Collapse Deformation

The failure modes of models 20-5, 20-10, 20-15 and 13-5 are annular collapse failure. The following paper takes 20-5 model and 13-5 model as examples to analyze the failure process of annular collapse failure combined with its pore pressure and soil pressure data.

#### 4.1.1. 20-5 Model

At these gentle slope angles, the effect of gravity alone could not cause visible movement of the soil layer. No obvious cracks were observed on the seabed surface after the small sliding caused by the angle adjustment. Therefore, pore pressure can accumulate in the sedimentary layer until deep fractures were formed in the clay and produced pockmarks. It can be seen from the Figure 7 that the sedimentary soil around the slope surface is still relatively intact. At these gentle slope angles, tensile failure was determined to be the major mechanism, resulting from the abrupt release of accumulated excess pore pressure. As the pore pressure is gradually released, the surface of the soil layer collapses.

To further elucidate the development process of slope failure, the 20-5 model is taken as an example to analyze the failure process of the slope according to the pore pressure measured at P1, P2, P3, and P4. Figure 15 shows the pore pressure development curve. After adjusting the angle of the model box, timing commences when the sensor outputs a stable hydrostatic pressure. As expected, the P1 sensor at the lowest position has the highest value, and the hydrostatic pressure is 3.27 kPa. The figure shows that the value of the pore pressure increases with the application of gas at the

beginning of the experiment; however, the value decreases rapidly after each increase, possibly because the accumulated gas diffuses outward and disrupts the surrounding soil. Similarly, the increase in the other two pore pressures also represents an alternating process of pore pressure accumulation and dissipation, reflecting the continued disruption of the sediment.

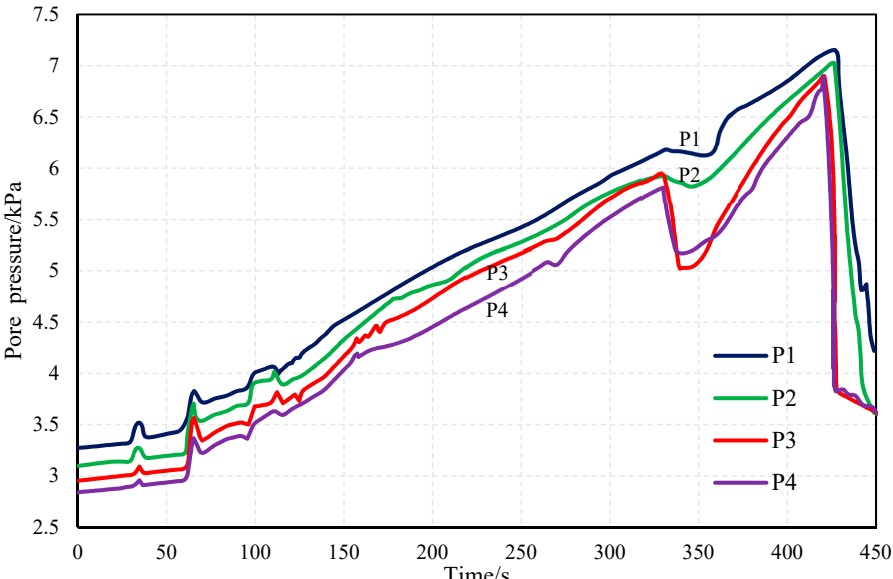

**Figure 15.** Measured pore pressure in Model 20-5.

The pore pressure tends to increase continuously, mainly due to the applied gas and its accumulation in the soil layer. The gas is enclosed in the soil layer and the pore pressure is increased because of the strong cementation of the silt and low permeability. The pore pressure drops suddenly until 330 s, especially at P3 and P4. The buried positions of these two sensors are just below the collapse of the slope (① in Figure 7d). It is established that the formation below the collapsed area may result in obvious damage. This is potentially because a crack is generated in the crucible, creating a path for the dissipation of the pore pressure. The slope caused major damage when the value of the pore pressure was finally reduced.

Figure 16 shows the variation of earth pressure data in the 20-5 model. At the beginning of the experiment, the earth pressure value did not change significantly, indicating that there was no obvious damage inside the soil layer. At the 300th second, the S4 data are reduced, which corresponds to the decrease of P3 and P4 in the pore pressure data. It may be that the seabed is damaged, which is supposed to be related to the occurrence of collapse deformation (① in Figure 7d). The values of S1, S2, and S4 are significantly reduced from the 360th second, which may be related to the occurrence of collapse deformation (② in Figure 7d). The deep soil layer produces cracks in the soil, and the soil particles in the cracks decrease with the amount of water vapor, causing soil pressure, and pore pressure to decrease. It can be speculated that the increase of the pore pressure may cause significant damage to the soil layer leading to the generation of cracks. The water-soil mixture rises up along the cracks and continuously washes the surrounding soil, causing damage to the soil structure and liquefaction of the seabed. As such, the soil pressure data are continuously reduced. As the soil layer continues to break down, pore pressure begins to dissipate. Eventually, owing to the continuous upward flow of soil particles, the surface of the seabed along the collapse-deformation generated split crack deposits.

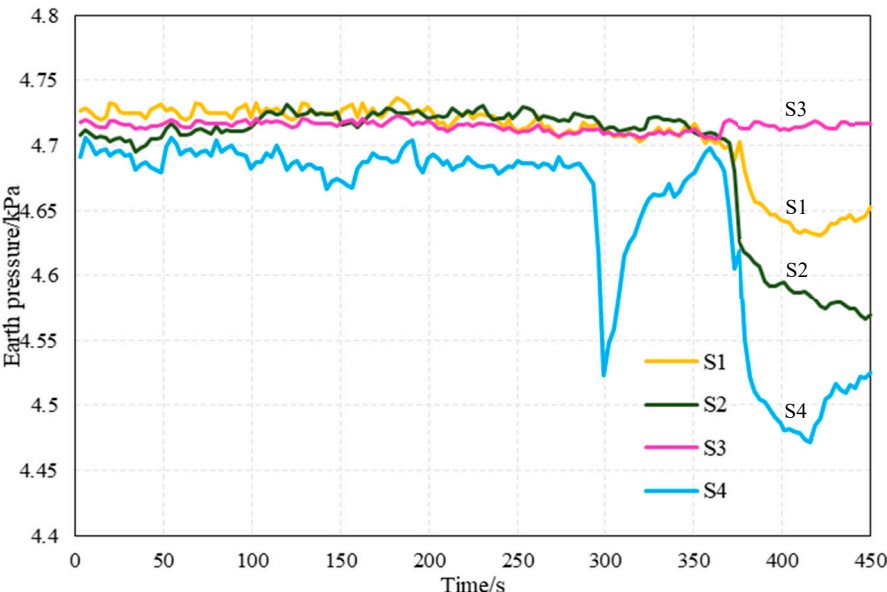

**Figure 16.** Measured soil pressure in Model 20-5.

### 4.1.2. 13-5 Model

Similar to the failure mode of a 20 cm thick seabed model, the 13-5 model is a collapse deformation similar to a pockmark. The effect of only gravity could not cause a visible movement of the soil layer on a gentle slope. As the gas accumulates and dissipates in the crack, the water and soil mixture oozes upward. Eventually, the surface of the seabed collapses and deforms similar to a pockmark.

Figure 17 shows the monitored values of the pore pressure for the 13-5 model. Similar to the 20-5 model, the hydrostatic pressure at P1 is the highest. Starting from the 35th second, several consecutive increases in the value of the pore pressure indicate the disruption of the deep region of the sedimentary soil. The cracks formed by these small damages lead to the dissipation of pore pressure and create conditions for generation of continuous cracks in the soil. The cumulative dissipation frequency of pore pressure is higher compared to the 20-5 model. This may be because the soil layer is thinner and more easily disrupted, which is not conducive to the accumulation of pore pressure. At approximately 245 s, the pore pressure drops abruptly, probably because a continuous crack inside the soil provides a channel for pore pressure dissipation. At approximately the 290th second, the slope is mainly disrupted, and the P1 pore pressure drops to the initial pore pressure value after failure.

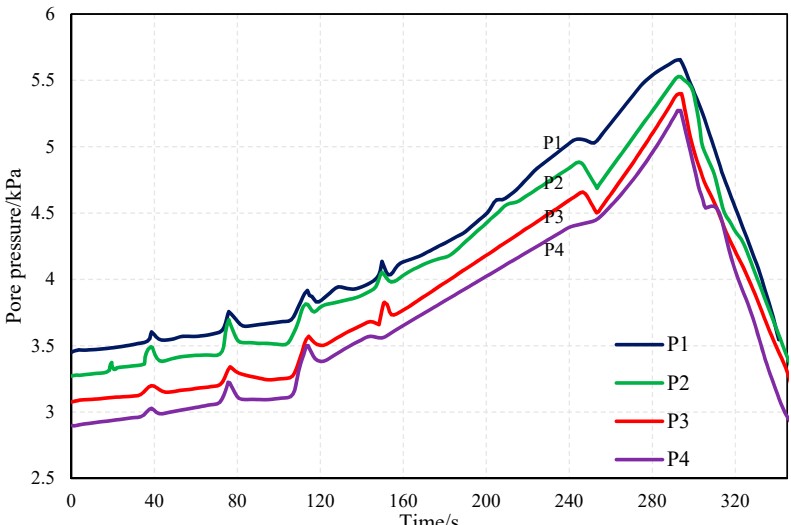

**Figure 17.** Measured pore pressure for Model 13-5.

Figure 18 shows the soil pressure monitoring data for the 13-5 model. The values of S3 and S4 change minimally, and the positions of the two sensors are not significantly disrupted. However, the values of S1 and S2 change significantly. At the beginning of the experiment, the value of the sensor gradually increased with a small amplitude and the soil gradually slid to the foot of the slope. At the 215th second, the sensor value at S1 drops significantly, probably because the internal disruption of the soil leads to collapse and provides a channel for pore pressure dissipation. At the 320th second, the earth pressure at S1 and S2 is significantly reduced because the main collapse deformation (② in Figure 10d) occurs at this time. The soil at the center of the deformation zone is in a liquefied state, and its effective stress is significantly reduced.

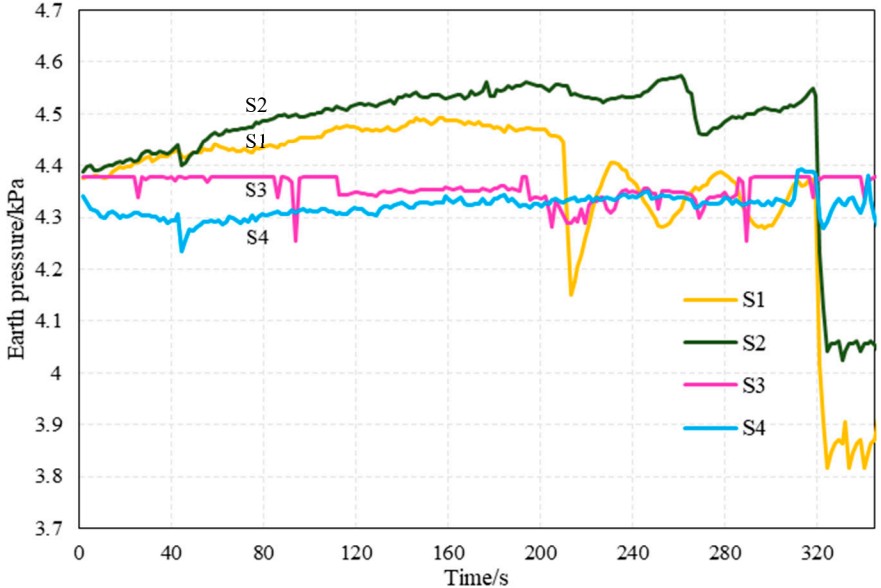

**Figure 18.** Measured soil pressure for Model 13-5.

In the 20-5, 20-10, 20-15, and 13-5 models, the collapse mode is collapse failure, similar to a pockmark. This is because the gentle seabed is conducive to the accumulation of pore pressure. When the pore pressure increases to a certain threshold point, cracks occur in the depth of the soil layer and the pore pressure dissipates along these cracks. When the soil particles stop moving, the pore pressure continues to accumulate at the original cracks. When the pressure accumulates to a certain threshold value, this leads to further extension and failure of the cracks. As the pore pressure accumulates and dissipates alternately, deep cracks in the formation continue to expand. Under the effect of excess pore pressure, the mixed fluid of water and soil transport the soil in the vicinity of the crack and seep upward through this region. This results in disruption of the soil structure and collapse deformation of the surface of the formation.

*4.2. Shear-Slip Failure*

The failure modes of model 13-10, 13-15, 8-5 and 8-10 are shear-slip failure. The following paper takes 13-15 model and 8-10 model as examples to analyze the failure process of shear-slip failure combined with its pore pressure and soil pressure data.

4.2.1. 13-8 Model

The 13-15 model is taken as an example to explain the mechanism of shear-slip failure. The relationship between pore pressure, earth pressure, and failure is discussed. Figure 19 shows the monitored data for the pore pressure. The hydrostatic pressure at P1 is 3.06 kPa. The pore pressure increased twice between the 30th and 60th seconds, indicating that this parameter exceeded the resistance of the overlying soil, thereby causing the expansion of the initial fracture. The pore pressure

then dissipated outward along the fracture. When the pore pressure decreases, the crack closes, and the gas continues to accumulate in the soil layer, causing a pore pressure increase. At the 155th second, the pore pressure began to decrease and the slope eventually failed. According to the change of pore pressure, the destruction process of the model may be as follows: In the 13-15 model, the initial crack caused by the initial slip of the slope continues to expand as the pore pressure accumulates and dissipates alternately. The water–soil mixture seeps upwards along the crack under the action of the excess pore pressure, continuously scouring both sides of the soil. This results in a weak sliding surface. The sediment layer slips along the sliding surface. When the sedimentary soil moves downward, the failed soil mixes thoroughly with water and is highly fluid. As such, slope failure is accompanied by dissipation of the pore pressure. Unlike the 20-5 and 13-5 models, the pore pressure cumulative extremes in the 13-15 model decrease. This may be because the seabed is more susceptible to damage due to the presence of initial fractures in the formation that are not conducive to the accumulation of pore pressure. In the 13-15 model, the initial slip of the slope caused the initial crack.

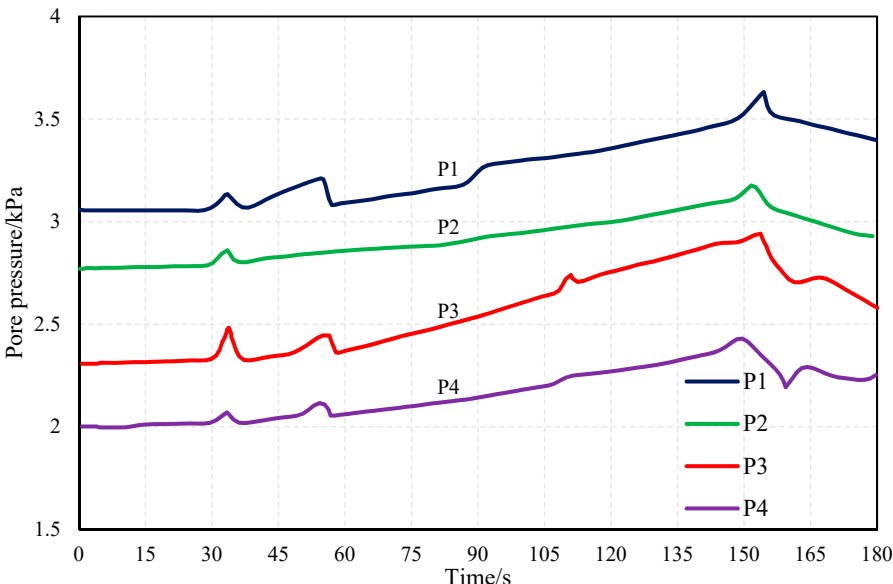

**Figure 19.** Measured pore pressure in Model 13-15.

Figure 20 shows the trend of earth pressure data in the 13-15 model. In the early stage of the experiment, the soil pressures at S1 and S2 increased slowly. The soil slowly descends along the slope, causing a gradually accumulation in this direction. Therefore, the soil pressure decreases greatly at S4. At the 160th second, the values of the three earth pressures decreased significantly, indicating that the slope eventually suffered shear–slip failure. Under the influence of excess pore pressure, the soil particles are highly mixed with water vapor, and the water–soil mixed fluid underwent a significant slip movement. This causes the soil structure to be disrupted, the effective stress to disappear, and the pore pressure to simultaneously dissipate.

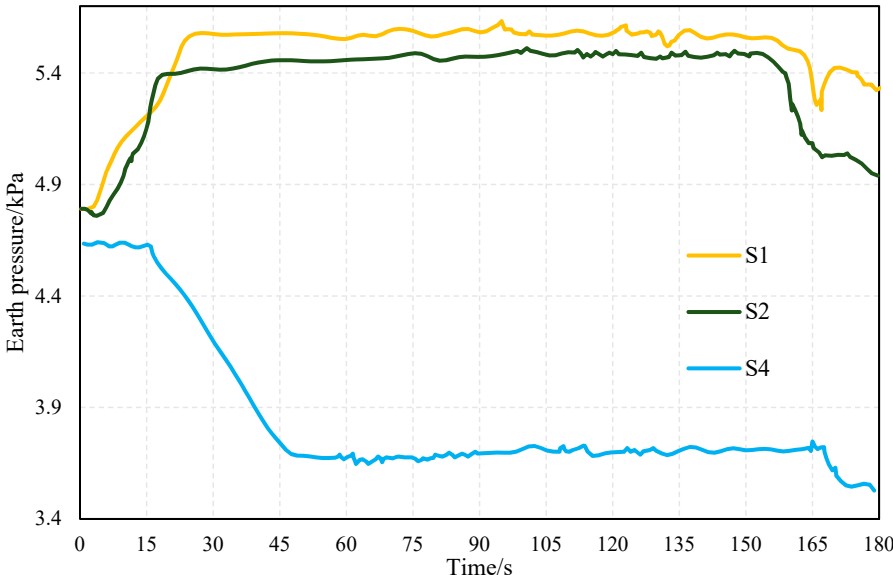

**Figure 20.** Measured soil pressure in Model 13-15.

### 4.2.2. 8-10 Model

The failure phenomenon and failure mode of model 8-10 are similar to model 13-15. Figure 21 shows the pore pressure measured at P1, P2, P3, and P4 for the 8-10 model. At the 40th second, the pore pressure at P1 is dissipated for the first time, indicating the occurrence of cracks in the soil layer. The crack provides a channel for gas diffusion, resulting in significant reduction in pore pressure. This creates adverse conditions for pressure accumulation. After 40 s, the pore pressure increases, and the slope is eventually destroyed. Unlike variation of the pore pressure in the previously analyzed model, the pore pressure accumulation time is significantly reduced in the 8-10 model. In thinner and steeper strata, gravity can cause slippage of the soil layer, and a reduced excess pore pressure can lead to total failure of the slopes. Therefore, the sedimentary strata fails earlier in the process.

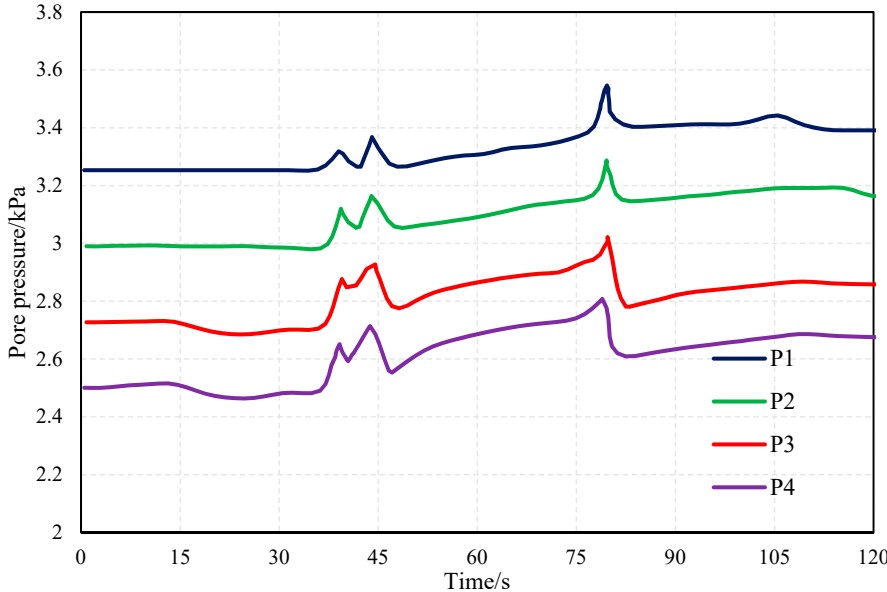

**Figure 21.** Measured pore pressure in Model 8-10.

Figure 22 shows that earth pressure is unchanged at the beginning of the experiment. At 40 s, the earth pressure at S1 and S2 increased significantly, while a decrease occurred at S3 and S4. At this

point, the slope may have slipped. In the final stage of the experiment, the earth pressure at S4 is again reduced to 3.79 kPa. When the formation was eventually destroyed, a large slippage occurred at the top of the slope.

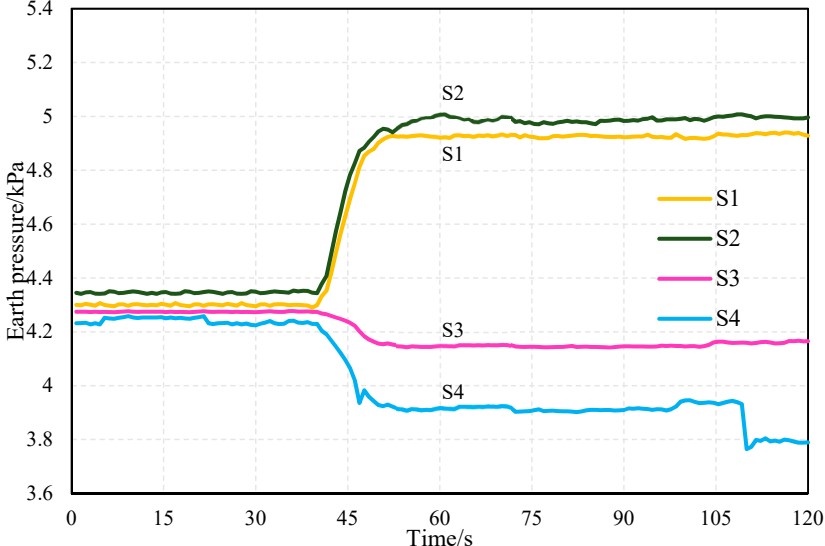

**Figure 22.** Measured soil pressure in Model 8-10.

In the 13-10, 13-15, 8-5, and 8-10 models, the collapse mode is shear–slip failure. In shallow or steep seabed, it is easy to produce tiny cracks, owing to the dual effects of gravity and excess pore pressure. These cracks provide channels for the release of pore pressure. Therefore, pore pressure is dissipated along these cracks. The sediment layer undergoes shear–slip damage along the sliding surface. This failure mode is similar to a submarine landslide.

## 5. Conclusions

This paper simulates the excess pore pressure caused by hydrate decomposition and observes the physical processes associated with the instability of the overlying seabed by a series of model tests, comparisons, and analyses performed on the evolution of slope failure under different formation conditions. The main experimental results are summarized as follows:

(1) Under the action of excess pore pressure caused by hydrate decomposition, the typical phenomenon of overlying seabed damage mainly includes pockmark deformation and shear–slip failure.

In shallower or steeper strata, shear-slip failure occurs in the slope. The existence of initial crack in the stratum is the main trigger cause. Under the action of the excess pore pressure, the shear strength of the soil at the crack is lost and the soil particles are displaced, it gradually develops into a weak sliding surface. Shear damage occurs along the sliding surface, accompanied by significant soil slip or debris flow.

In thicker formations or gentler slopes, the surface of the seabed has a collapse deformation feature. The occurrence of cracks in the deep soil layer is the main failure mechanism. When the excess pore pressure develops beyond the tensile strength of the soil, the vertical displacement of the soil particles occurs, and cracks are generated inside the soil. The mixture of water and soil seeps up along the crack, causing damage to the surrounding soil structure, which in turn causes collapse and deformation of the sediments.

(2) The thickness and slope of the seabed, among other factors, affect the type and extent of seabed damage. The slope has a higher influence on the stability of the formation than the thickness.

**Author Contributions:** Conceptualization, T.L.; Formal analysis, Y.L. and L.Z.; Methodology, L.Z. and L.G.; Software, X.Y.; Validation, X.Y.; Writing—original draft, Y.L.; Writing—review & editing, T.L. and L.G.

**Funding:** This research was funded by the National Natural Science Foundation of China under Grant 41672272 and 41427803; Primary Research and Development Plan of Shandong Province under Grant 2017GGX30125; National Natural Science Foundation of China under contract Nos. 41806075; Marine Geological Process and Environmental Function Laboratory Open Fund Grant MGQNLM-KF201709 and The Fundamental Research Funds of Shandong University.

**Acknowledgments:** We appreciate anonymous reviewers who gave comments to revise the paper.

**Conflicts of Interest:** The authors declare no conflict of interest.

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
