# Peer review of "Experiment and Analysis of Submarine Landslide Model Caused by Elevated Pore Pressure"

_jmse, doi:10.3390/jmse7050146_

Reviewer 1 Report

The manuscript titled “Experiment and analysis of submarine landslide model caused by elevated
pore pressure” by Tao et al. is focused towards understanding the effect of elevated pore pressure in gas hydrate decomposition. This manuscript presents the impact of the thickness of sediment and inclination of slope in relation to elevated pore pressure and gas hydrate decomposition using specific experimental conditions (stated in the manuscript) as a guide. Although the purpose of the paper is worthwhile a lot of important details were left out. Outlined below are some of the suggestions that will possibly improve the manuscript:

ABSTRACT
The abstract seems to be vague and not well written. The abstract for this manuscript needs to be greatly improved in such a way that there is a flow and the purpose of the paper is well related and communicated.
Present the rationale of investigating the topic and integrate the problem to introduce the goal of the study.
Report major findings.
Present interpretation of results.
Layout the implication of results concerning study objectives
All the above needs to be well linked and presented in the abstract.

INTRODUCTION
There is a need to strengthen and improve the introduction section of the manuscript.
The line “The gas hydrate is a special ice-like….” Special ice-like? What makes it special? Gas hydrate should be better defined and explained.
Include the below reference at the end of the first paragraph in Introduction.
Subbarao Yelisetti, George D. Spence, Michael Riedel (2014), Role of gas hydrates in slope failure on frontal ridge of northern Cascadia margin, Geophysical Journal International, 199(1), 441-458, doi:10.1093/gji/ggu254.
On line 3 of the second paragraph in Introduction: …..decrease of sediment strength which in turn leads to slope failure (Yelisetti et al., 2014).
Factors that may affect or trigger destabilization of gas hydrates should be discussed explicitly in the introduction.
Poor transitioning within sentences.
The main purpose of the paper needs to be expanded and introduced to integrate it to the identified gap appropriately.
4th paragraph under Introduction: On the line “ Current research indicates the main mechanism of seabed instability is the high pore pressure caused by hydrate decomposition”there is no citation. These studies need to be cited. Add the following after this line.
Yelisetti et al (2014) observed that the contrast in sediment strength between hydrated and non-hydrated sediments acts as a glide plane for failure on the northern Cascadia margin.
There are several studies of pore pressure elevation this should be presented so that the scope and stage at which research is in this area can be well played out and known.

METHODOLOGY
The line “According to the dimensionless analysis of seabed damage caused by hydrate decomposition” seems to be out of place and vague.
On page 10 “start experiment” is written as an instruction rather than a procedure.
Why was the silty soil of the Yellow River Delta beach used? Was there any rationale for using this? If so, this should be presented to help justify the scope of this research.

RESULTS
Result section should be separate from the methodology.
Results need to be written out before discussing these. This will help the presentation of the paper.
Delete Chinese letters on figure 3-14

DISCUSSION
Discussion section should be separate from the methodology.

CONCLUSION
What is the implication of the major findings? Implication of the research should be presented in conclusion.

Author Response

Responses to Reviewers and Editor Comments

Date: 21 April 2019

Title: Experiment and analysis of submarine landslide model caused by elevated pore pressure

First of all, we thank the Editor and the reviewers for their constructive comments and suggestions. A point to point reply to the comments of the reviewer are given as follows:

1. ABSTRACT

The abstract seems to be vague and not well written. The abstract for this manuscript needs to be greatly improved in such a way that there is a flow and the purpose of the paper is well related and communicated.

Present the rationale of investigating the topic and integrate the problem to introduce the goal of the study.

Report major findings.

Present interpretation of results.

Layout the implication of results concerning study objectives

All the above needs to be well linked and presented in the abstract.

Reply: According to your opinion, we reorganized the abstract and added rationale of investigating the topic, rationale of investigating the topic and major findings in page 2.

2. The line “The gas hydrate is a special ice-like….” Special ice-like? What makes it special? Gas hydrate should be better defined and explained.

Reply: We explained the particularity of gas hydrate and highlighted in red in the manuscript, see page 2, lines 5.

3. Include the below reference at the end of the first paragraph in Introduction.

Subbarao Yelisetti, George D. Spence, Michael Riedel (2014), Role of gas hydrates in slope failure on frontal ridge of northern Cascadia margin, Geophysical Journal International, 199(1), 441-458, doi:10.1093/gji/ggu254.

Reply: We added the reference at the end of “This compound is not only a natural source of clean energy but also a potential factor that induces geological hazards” and highlighted in red in the manuscript, see page 3, lines 3-4.

4. On line 3 of the second paragraph in Introduction: …..decrease of sediment strength which in turn leads to slope failure (Yelisetti et al., 2014).

Factors that may affect or trigger destabilization of gas hydrates should be discussed explicitly in the introduction.

Reply: We explained the trigger destabilization of gas hydrates. We highlighted in red in the manuscript, see page 2, lines 2-4.

5. Poor transitioning within sentences.

Reply: We have professionally polished the manuscript throughout.

6. The main purpose of the paper needs to be expanded and introduced to integrate it to the identified gap appropriately.

Reply: We expanded the main purpose of the paper and highlighted in red in the manuscript, see page 4, the last paragraph.

7. 4th paragraph under Introduction: On the line “Current research indicates the main mechanism of seabed instability is the high pore pressure caused by hydrate decomposition”there is no citation. These studies need to be cited. Add the following after this line.

Yelisetti et al (2014) observed that the contrast in sediment strength between hydrated and non-hydrated sediments acts as a glide plane for failure on the northern Cascadia margin.

Reply: We have added the citation and highlighted in red in the manuscript, see page 3, lines 16-18

8. The line “According to the dimensionless analysis of seabed damage caused by hydrate decomposition” seems to be out of place and vague.

Reply: We have deleted this sentence.

9. On page 10 “start experiment” is written as an instruction rather than a procedure.

Reply: We have changed “start experiment” to “experimental procedure”, and highlighted in red in the manuscript, see page 8, lines 13

10. Why was the silty soil of the Yellow River Delta beach used? Was there any rationale for using this? If so, this should be presented to help justify the scope of this research.

Reply: The gas hydrate reservoir has low permeability. Silt is conducive to the accumulation of excess pore pressure. Zhang et al. established an empirical strength model after triaxial tests of methane hydrate and glacial silt. Zhang and Wu found that there were differences in the formation and decomposition of hydrate in different soils. Therefore, we selected silt as the soil sample for this experiment.

11. Result section should be separate from the methodology.

Results need to be written out before discussing these. This will help the presentation of the paper.

Reply: We have rewritten the third section. We split the third section into results and discussion

12. Delete Chinese letters on figure 3-14.

Reply: We have redrawn all the figures.

13. Discussion section should be separate from the methodology..

Reply: We have rewritten the third section. We split the third section into results and discussion.

14. What is the implication of the major findings? Implication of the research should be presented in conclusion.

Reply: According to your opinion, we rewrite the conclusion section. We listed two major findings, and added the Implication of the research, see page 28-29.

Reviewer 2 Report

Review :
Experiment and analysis of submarine landslide model
caused by elevated pore pressure

While this is on the face of it a very simple set of experiments, thus should not necessarily be considered a detraction. These simple experiments do seem to replicate key features in the sub-surface marine cases, and for the purposes of creating slips, there is indeed no obvious need for the disturbing gas to be methane or CO2.

It is not clear how these thin layers of soil are related to a multi-meter thick true marine environment, however.

But Table 3.1 does indeed seem to list known sub-sea slope failure modes.

One question : what happens to all the soil in the experiment that has slipped ?
In figure 3-10 for instance, there appears to be a net loss of most of the soil.

Thorough proof-reading is recommended, including the following :

P4 : Confusing start to this paragraph :

Experimental introduction
According to the dimensionless analysis of seabed damage caused by hydrate
decomposition, In this paper,

P15 : repeated sentence :
In order to study the effect
of slope gradient on slope stability, the slopes of the silt layers were set to 5°, 10° and
15° in each of the three models.

P15 : Incomplete sentence :
As shown in Figure 3-10, due to the larger

angle of the slope, the top o

P22 –Fig. 3-13 – P23 Figs. 3-15 Graphs not all in English.

Recommendation, Publication with minor corrections,

Author Response

Responses to Reviewers and Editor Comments

Date: 21 April 2019

Title: Experiment and analysis of submarine landslide model caused by elevated pore pressure

First of all, we thank the Editor and the reviewers for their constructive comments and suggestions. A point to point reply to the comments of the reviewer are given as follows:

1. P4: Confusing start to this paragraph:

Experimental introduction

According to the dimensionless analysis of seabed damage caused by hydrate

decomposition, In this paper,

Reply: We have deleted this sentence.

2. P15: repeated sentence:

In order to study the effect

of slope gradient on slope stability, the slopes of the silt layers were set to 5°, 10° and

15° in each of the three models.

Reply: We have deleted the repeated sentence.

3. Incomplete sentence:

As shown in Figure 3-10, due to the larger

angle of the slope, the top o

Reply: We have reviewed the manuscript and to avoid such mistakes.

4. P22 –Fig. 3-13 – P23 Figs. 3-15 Graphs not all in English

Reply: We have redrawn all the figures.

Round  2

Reviewer 1 Report

Check P3, line 3. Delete Error! Reference source not found.

The heading “Experimental procedure appears twice”. Delete one or change it to something else.

In P10, change “(5)After the experimentation process” to something else.

In P10, change “The resultsof experimental” to something else.

In P21 and P24, delete Chinese characters

You do not need summary and conclusions. Change the summary heading in P26 to something else.

Conclusion seem to be too long.

Author Response

Responses to Reviewers and Editor Comments Date: 25 April 2019 Title: Experiment and analysis of submarine landslide model caused by elevated pore pressure First of all, we thank the reviewer for your constructive comments and suggestions. A point to point reply to the comments of the reviewer are given as follows: 1. Check P3, line 3. Delete Error! Reference source not found. Reply: We have found the right reference and highlighted in red in the manuscript, see page 3, lines 1. 2. The heading “Experimental procedure appears twice”. Delete one or change it to something else. Reply: we have changed the first “Experimental procedure” to “Description of model tests” and highlighted in red in the manuscript, see page 8, lines 2. 3. In P10, change “(5) After the experimentation process” to something else. Reply: We also feel that this sentence is inappropriate and have delated it. 4. In P10, change “The results of experimental” to something else. Reply: We have changed “The results of experimental” to “Experimental results and phenomena”, and highlighted in red in the manuscript, see page 10, lines 1. 5. In P21 and P24, delete Chinese characters Reply: We have deleted Chinese characters in page 21 and 24. 6. You do not need summary and conclusions. Change the summary heading in P26 to something else. Reply: We have deleted the summary in page 22 and 26. 7. Conclusion seem to be too long. Reply: According to your opinion, we have rewritten and deleted the wordy content in the conclusion section in page 28.
